# Meta-analysis reveals the predictable dynamic development of the gut microbiota in commercial pigs

Wenxuan Dong,[1] Nicole Ricker,[2] Devin B. Holman,[3] Timothy A. Johnson[1]

**ABSTRACT** Understanding the mechanisms of microbiome assembly during host development is crucial for successful modulation of the gut microbiome to improve host health and growth. However, results from previous microbial intervention studies to improve swine growth have largely been inconsistent due to the constantly changing nature of the gut microbiota and limited longitudinal sampling. Detailed characterization of the swine gut microbiota through meta-analysis allows us to understand the dynamics of microbial community succession, as well as the transient and natural variations between time points and animals. A total of 3,313 fecal microbial communities from over 349 pigs covering 60 time points (from birth to market age) from 14 publications were included in this meta-analysis. Despite differences in animal breeds and management, generalizable patterns of community assembly were identified. Alpha diversity continuously increased during early stages of animal growth and more slowly in the later stages. Beta regression analysis revealed that more microbial taxa were enriched, while fewer were excluded by the gut microbiota. The microbial community structure also changed significantly between days at early ages and became more similar as pigs aged, as revealed by dissimilarity and distance metrics. Dirichlet multinomial mixtures analysis supported gradient microbial clustering in longitudinal pig fecal samples, and we found that the early samples spread to more clusters than those from older pigs. Random forest regression identified 30 operational taxonomic units as potential microbial biomarkers for modeling swine gut microbiota development. External validation showed this model could be generalized to future microbiome studies conducted in suckling and weaning pigs and adds to the toolkit to quantify community succession.

**IMPORTANCE** The swine gut microbiome undergoes an age-dependent assembly pattern with a developmental phase at early ages and a stabilization phase at later ages. Shorter time intervals and a wider range of data sources provided a clearer understanding of the gut microbiota colonization and succession and their associations with pig growth and development. The rapidly changing microbiota of suckling and weaning pigs implies potential time targets for growth and health regulation through gut microbiota manipulation. Since swine gut microbiota development is predictable, swine microbiota age can be calculated and compared between animal treatment groups rather than relying only on static time-matched comparisons.

**KEYWORDS** swine gut microbiome, longitudinal, dynamic development, microbiota age, community succession

T he intestinal microbiota plays a vital role in both human and animal health. Mounting evidence has proven the associations between the human gut microbiota and immunity, metabolism, and health (1–4). Dysbiosis of the intestinal microbial community may cause various immune and metabolic-associated diseases (4–6). Likewise, in food animals, gut microbial variations may also explain to some extent

Address correspondence to Timothy A. Johnson, john2185@purdue.edu.

The authors declare no conflict of interest.

See the funding table on p. 13.

variations in feed efficiency, post-weaning disease, and growth performance (7–11). Although probiotics and/or prebiotics are promising strategies for modulating the gut microbiome to control pathogens and promote growth in swine production (12), these agents are still not as reliable as antibiotics (13–15). A detailed understanding of how the healthy and normal pig gut microbiome develops will likely aid in the development and successful use of probiotics and prebiotics as antibiotic alternatives.

The longitudinal composition of the swine microbiota has most commonly been described at weekly or monthly intervals (16). Le Sciellour et al. collected fecal samples from pigs at 52, 99, 119, 140, and 154 days of age and found that post-weaning, the most significant change occurred in the microbiota from 52 to 99 days, although consistent community succession was observed during the entire finishing period (17). Han et al. studied the dynamic change in the fecal microbiota of 32 healthy pigs using five time points (10, 21, 63, 93, and 147 days of age) across three growth stages and concluded that the greatest variability occurred during the weaning phase with increased stability as the pigs reached market weight (18). In another experiment, a total of 69 core microbes belonging to 19 phyla were identified along with 91 stage-specific taxa, but at a coarse time resolution with approximately 10-day intervals from birth to market (19, 20). In terms of early-life microbial development, daily sampling of healthy pigs during the first week of life revealed the age-dependent patterns of the pig gut microbiota, (21) and in another study, phylogenetic diversity increased during nursing and reached a plateau 2 weeks after weaning (22).

Similarly, longitudinal studies have also determined the compositional and functional development of the human infant microbiota and have provided a foundation for mechanistic investigations of microbes and host health (23). However, a clearer view of the development of the human gut microbiota was revealed using fecal samples obtained on a near-daily basis (16, 24). The importance of high-frequency sampling over an extended time period has been emphasized to ensure accurate interpretation of community succession and host-microbe interactions (16, 25, 26).

Thus far, studies of the community succession of the swine microbiota have only used large sampling time intervals, which could lead to inconsistencies between studies. High-resolution analysis with shorter time intervals is critical in determining the continual dynamics of microbiota composition rather than infrequent "snapshot" samples (16, 24). Differentially abundant taxa identified during snapshot sampling may reflect natural community dynamics, rather than the selection of taxa by a particular treatment (27–29). Moreover, swine gut microbiome research requires a detailed description of microbial succession in the gut to serve as a benchmark by combining and comparing different pig gut microbiome studies to identify potential microbial biomarkers (30–32).

In the current study, we performed an individual participant data (IPD) meta-analysis of fecal samples collected from commercial pigs in order to fill the knowledge gap that exists on the detailed succession of the swine gut microbiota. Overall, the swine gut microbial community develops in a consistent and predictable manner until a stable community is established. We also identified and utilized age-discriminant taxa to predict the microbiota maturation state and identified core microbial members that persist in the pig gut.

## RESULTS

### Meta-analysis characteristics

Data from 14 studies published from 2019 to 2021 were included in this meta-analysis (Fig. 1A). There were 3,313 fecal samples (stools or swabs) collected from 60 time points (0–183 days of age) from more than 349 pigs. Overall, 16S rRNA gene sequences were available from four continents and six countries (Fig. 1B). For longitudinal investigations, different batches of DNA extraction kit reagents might be a substantial source of variance (33). DNA extraction kits produced by Qiagen were the most widely used for swine gut microbial studies (9 out of 14 studies, Table S1). The choice of 16S rRNA gene

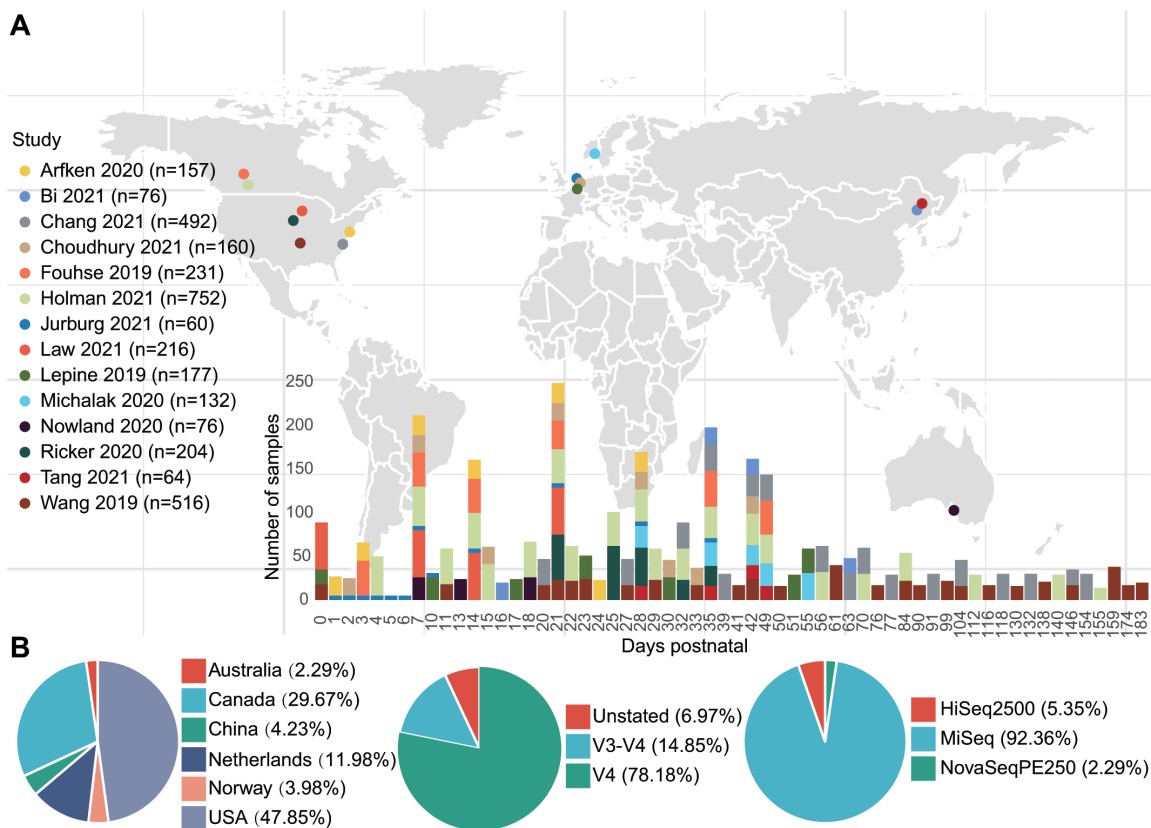

FIG 1 A schematic overview of the current meta-analysis. (A) The map depicts the locations from which samples were collected and the number of samples from each study. The map was created in R with the ggmap package. (B) The bar plot shows the sample distribution over pig age and study. The pie charts show the data distribution by country, hypervariable region sequenced, and sequence platform used. Additional details can be found in Table S1.

region(s) and sequence platforms also influence the bacteria that can be identified (34). In the current study, we found that most of the sequence data were generated from the V4 region (78.18% of samples) and sequenced using the Illumina MiSeq platform (92.36% of samples). Only 649 samples (18.55% of all samples) were collected from growing and finishing pigs, indicating a microbial data deficiency in older pigs and also indicating that most microbial manipulation studies were focused on young pigs.

## Alpha diversity and taxonomic composition

Generally, the age of the pigs was positively correlated with alpha diversity. Faith's phylogenetic diversity index, which estimates the community phylogenetic diversity, increased over a large time span (Spearman's rho = 0.76, $P < 0.001$; Fig. 2A). Similar increasing trends were also observed for Pielou's evenness, the number of observed operational taxonomic units (OTUs), and the Shannon diversity index (Fig S2A through C). Interestingly, in the meconium (day 0), there was a high level of microbial diversity, as measured by Faith's phylogenetic diversity and the number of observed OTUs, which were also highlighted in the original published paper (20) (Table S2). We also observed that time (age) exerts a stronger effect on alpha diversity than dietary treatments or fecal microbiota transplantation (Table S3; Fig. S3 and S4).

We next sought to characterize the compositional variations with the passage of time by identifying the most relatively abundant phyla and genera, as well as taxonomic composition at the phylum and genus levels (Fig. 2B; Fig. S5A). We used beta regression to model how each phylum or genus increased or decreased in relative abundance over time (35). Of the 20 most abundant phyla in the gut microbiota, the temporal distribution of 14 phyla differed significantly from birth to market age (Fig. 2C). There were

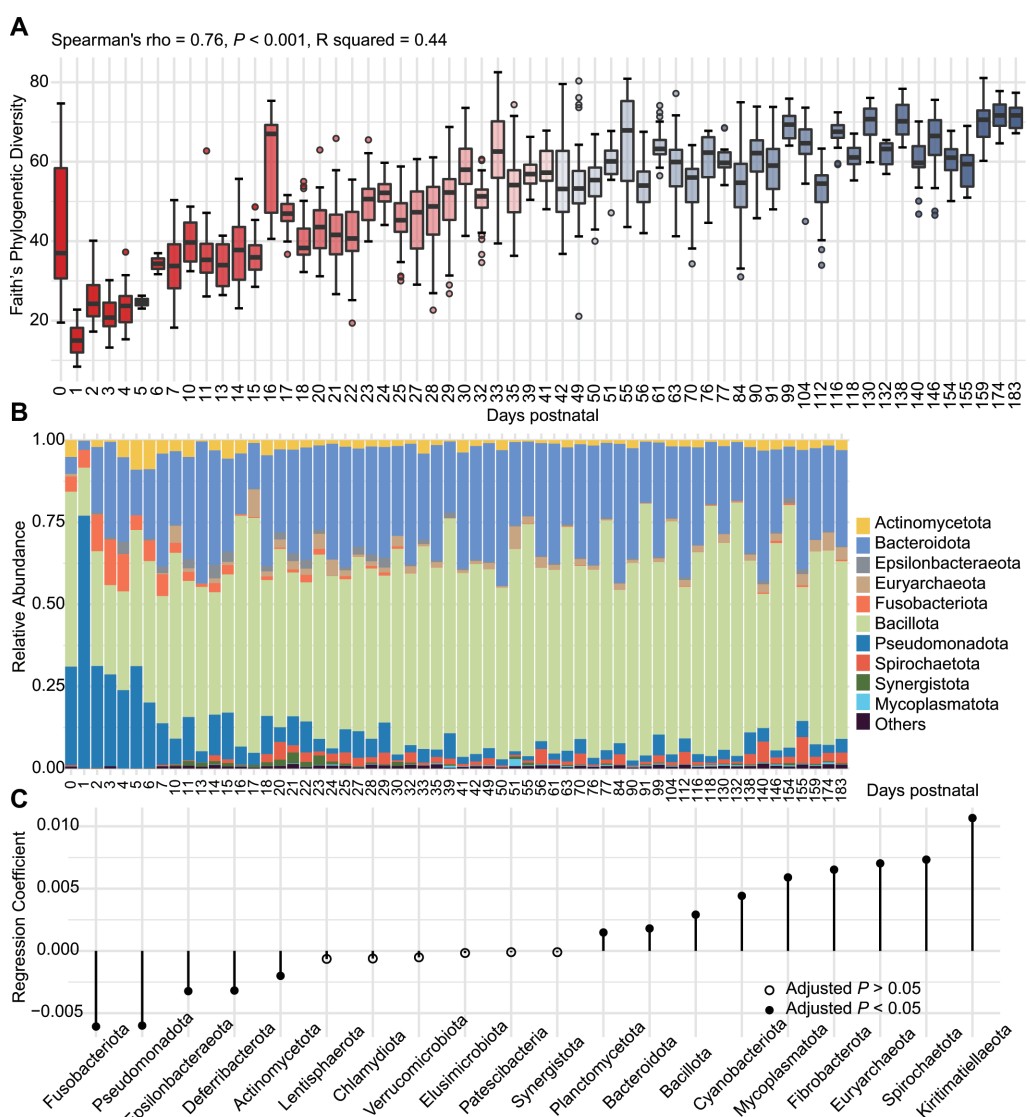

**FIG 2** Temporal development of the swine gut microbiota, as measured by diversity and composition. (A) Faith's phylogenetic diversity in the swine gut microbiota from birth through to marketing. Spearman's rho shows the positive correlation between Faith's phylogenetic diversity and pig age. Statistical comparisons can be found in Table S2. (B) Bar plots of the relative abundance of the 10 most relatively abundant phyla over the course of time. Each bar represents the average value of the control samples from each time point. (C) Beta regression coefficients indicate phyla that either increased (>0) or decreased (<0) in relative abundance over time.

nine microbial phyla whose relative abundance significantly increased over time and five phyla that decreased. Similarly, of the 62 most relatively abundant genera (>0.3% relative abundance), there were 36 genera whose relative abundance significantly increased over the course of time, while 21 genera decreased (Fig. S5B). The absolute values of the regression coefficients were greater for the time-enriched taxa than the time-depleted taxa, implying that more microbial members are enriched by the pig gut over time.

## The stabilization pattern as revealed by beta diversity

To visualize the change in microbial community structure over time and to identify potential driving forces that explain variation in the current data sets, we used principal coordinate analysis (PCoA) ordination plots in combination with permutational multivariable analysis of variance (PERMANOVA) and permutational multivariate analysis

of dispersion (PERMDISP) on beta diversity distances between control samples from untreated pigs. We first assessed sources of heterogeneity using the variance (PER-MANOVA $R^2$ value) explained by pig age, growth stage, study, hypervariable region sequenced, sequencing platform, and country of origin. Not surprisingly, the two most influential factors affecting the swine gut microbial structure were "study" and the growth stage of pigs (Fig. 3A and B; Table S4; Fig. S6A and B). We next inspected the other factors that shape the gut microbiota by dividing the samples into the suckling, weaning day, weaning, growing, and finishing periods. Weaning day was used as its own period based on the assumption that the microbiota composition on the day of weaning is more similar in all pigs because they are well adapted to the sow's milk. We found the variables that explained the largest amount of the variance were (in order of importance) study, 16S rRNA gene hypervariable region sequenced, and "age." The importance of the

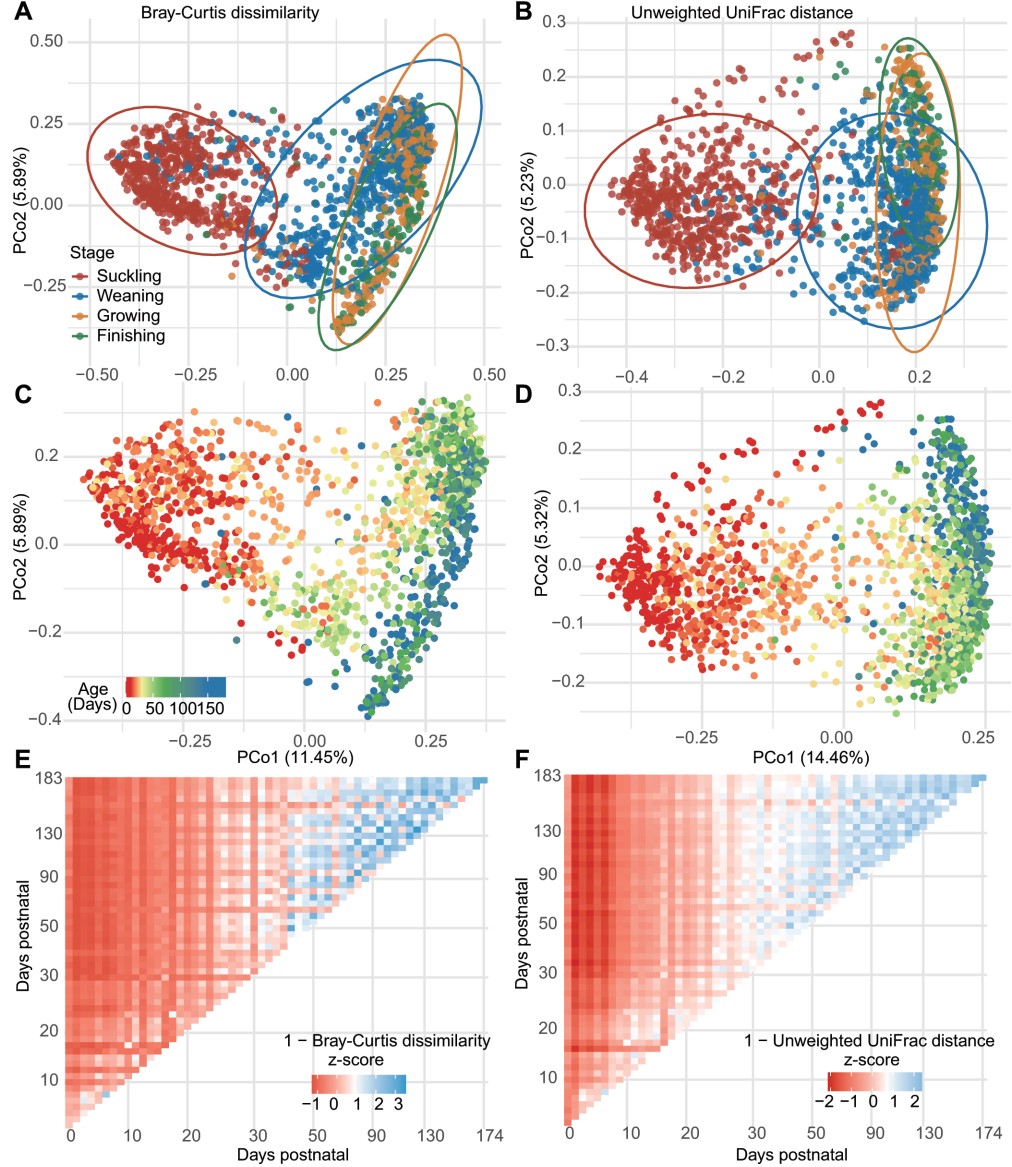

**FIG 3** The gut microbiota beta diversity stabilizes 40–50 days after birth. (A–D) Principal coordinate analysis plot of the dissimilarities or distances between samples from non-treated pigs colored by growth stages (A and B) and growth ages (C and D). (E and F) Heatmaps depicting the mean pairwise z-score for similarity between control samples from untreated pigs from each time point, calculated as (1 − beta dissimilarity). Bray-Curtis dissimilarity was used in panels A, C, and E. Unweighted UniFrac distance was used in panels B, D, and F. All statistical comparisons are presented in Table S3.

"country" factor is confounded with study and thus not considered on its own (Table S4; Fig. S7).

We next measured the effect of pig age on the gut microbiota. This effect was observable across the first principal coordinate of the PCoA plots and was consistent across the four different types of dissimilarity and distance measures (Fig. 3C and D; Fig. S6C and D). Likewise, the amount of variation partitioned to pig age was comparable among the four dissimilarity and distance measures (Table S4). We also found that the gut microbiota shifted over the first 30–50 days post-natal but stabilized thereafter regardless of dissimilarity or distance measure used (Fig. 3E and F; Fig. S6E and F). Overall, these data suggest that the swine gut microbiota is highly variable during early life and stabilizes upon entry into the growing period.

## Age-specific community clusters

Next, we tested whether the pig gut microbiota clustered into different community types according to age using the Dirichlet multinomial mixtures (DMM) model, including both control and treated pigs from each study (36). The DMM model gave a different optimal cluster number for different input OTU tables: 10 clusters for the OTU table with a rarefaction depth of 7,550 sequences, 16 clusters for the raw OTU table after removing samples with total counts lower than 5,000 and 31 clusters for a collapsed OTU table at the genus level (Table S5). We visualized the distribution of the 10 clusters formed over time from the rarefied OTU table and found three distinct phases of microbiota succession: a developmental phase (days 0–17, 4 dominant clusters), a transition phase (days 18–63, 9 dominant clusters), and a stable phase (days 70–183, 4 dominant clusters) (Fig. 4A).

To circumvent the heterogeneity associated with including 14 independent studies, we also analyzed separately the three largest studies that had samples from all growth stages and found similar cluster distribution patterns (Fig. S8A) (5, 20, 37). A clear growth stage-specific progression was also observed when restricting the analysis to two sequential growth stages at a time, with more clusters present in the early growth stages and fewer community types clustering at the end stages (Fig. S8B through D). Various animal treatments (fecal microbiota transplant, antibiotic treatment, and feed restriction) were equally represented in the DMM clusters, so we concluded that age was the primary factor in community clustering patterns (Fig. S8E). Next, we identified the 10 OTUs that contributed the most to each cluster and compared the relative abundance of the top five taxa within each cluster against all other clusters (Fig. 4B; Tables S6 and 7). As expected, a general increasing trend in alpha diversity was also observed in DMM clusters over the chronological order of their dominance (Fig. 4C; Fig. S8F through H). The highest diversity was observed in the transition phase dominant cluster F, which could be explained by the continuous recruitment of the new microorganisms, while early colonizers were not completely lost from that cluster (Table S8).

## Pig age is predictable by the gut microbiota

After identifying clusters and the greatest contributing taxa within each cluster, we next sought to identify OTUs that could be used to discriminate pig age. We identified 30 OTUs correlated with pig age in the gut using random forest (RF) models (Fig. 5A; Table S9). Our models explained 95.52% of the variance related to pig age. All microbial taxa were ranked in order of importance based on their contribution to the marginal increase in mean squared error to the model when their relative abundance values were permuted. Consistent with the results from the DMM clustering, we found that some OTUs were important for both age discrimination and DMM clustering differentiation (Fig. 5B).

Next, we applied our models to the training, test, and external validation data sets to see the extent to which our RF models could be generalized to samples not used to generate the model (Fig. 5C). We found that our models performed better with the training (Pearson' $r = 0.99$, $R^2 = 0.99$) and test (Pearson' $r = 0.98$, $R^2 = 0.96$) data sets

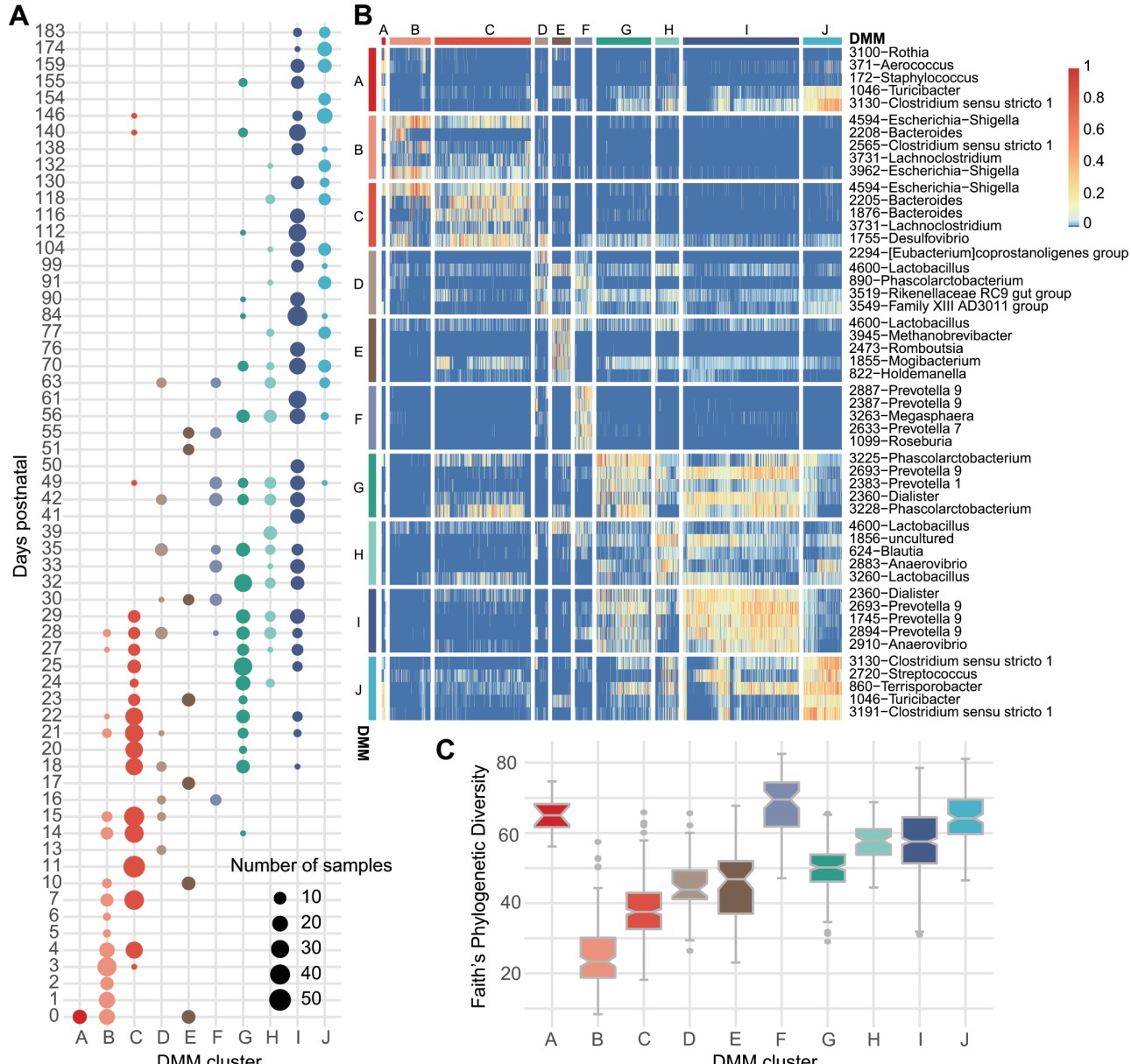

**FIG 4** The gut microbial community development as measured by age-specific Dirichlet multinomial mixtures (DMM) clusters. (A) Distribution of samples from untreated pigs in the identified 10 DMM clusters (x axis) using the DMM model at each time point (y axis). The size of each circle represents the number of samples within each cluster at each time point. (B) Heatmap showing the five OTUs that contributed the most to each cluster (rows) and their relative abundance within each sample (columns). Samples are sorted by cluster assignment. (C) Box plots of Faith's phylogenetic diversity within each DMM cluster. Boxes show median and interquartile ranges, whiskers show ±1.5 IQR from the quartiles. Statistics in Table S8.

than the external validation (Pearson' $r = 0.71$, $R^2 = 0.51$), although there were significant ($P < 0.001$) linear correlations between the chronological age and he microbiota age observed in all three data sets. Based on the data distribution, we further split our external validation data set into early samples (age <80 days) and late samples (age >80 days), and the predictive performance was much improved in the early (Pearson' $r = 0.79$, $R^2 = 0.62$, $P < 0.001$) compared to the late (Pearson' $r = -0.31$, $R^2 = 0.10$, $P = 0.006$) samples (Fig. S9). More robust predictive performance was also observed for early samples in training and test data sets, indicating that samples from early stages contributed more global variation in the entire data set (Table S10). Moreover, based on Pearson's correlation coefficient and the accuracy of the range of predicted microbiota

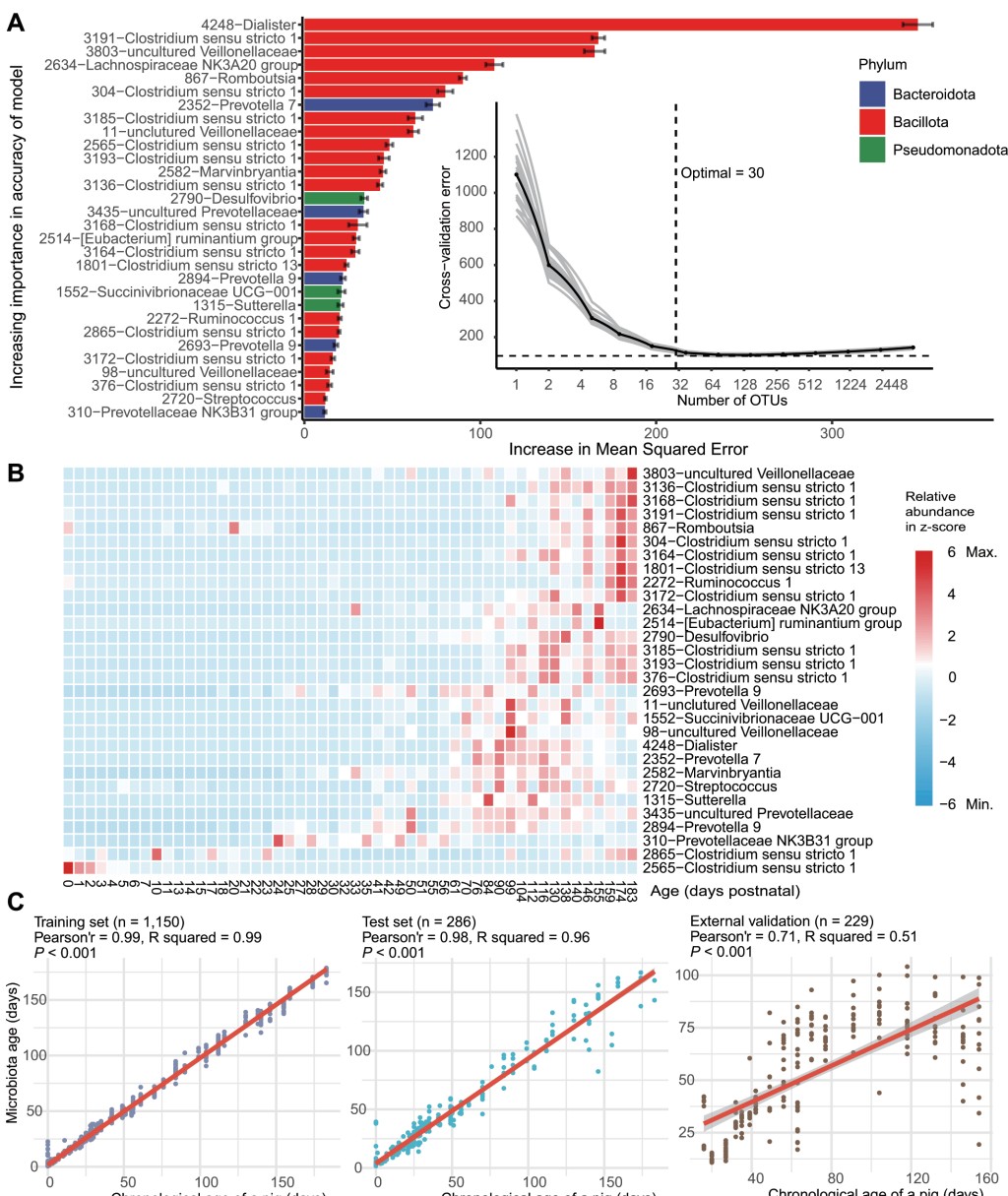

FIG 5 microbial taxonomic biomarkers for gut microbiota maturation in samples from untreated pigs. (A) Thirty age-discriminant OTUs were discovered by regressing their relative abundances against the chronological ages using the random forest algorithm with 20 iterations, listed in descending order of importance to the model's accuracy. Taxonomic annotations and representative sequences can be seen in Table S9. The percentage of increase in mean squared error of the models when the relative abundance of each OTU was permuted was used to assess the importance. The insert represents the 10-fold cross-validation error used to determine the optimum number of input OTUs to regress the age in the model. Gray lines represent each iteration, and the black line represents the mean values. (B) Heatmap showing the relative abundance of the top 30 age-discriminant microbial taxa against the chronological age of the control samples from untreated pigs used to train the models. The relative abundance z-score indicates enrichment (positive z-score, red) or depletion (negative z-score, blue) of a particular microbial taxon over the course of time. (C) Microbiota age predictions in the data sets used to train the models (training set), the data sets excluded for training the models but from the same studies as the training set (test set), and the data sets which are independent from training and test sets and derived from different studies (external validation).

ages, models created using all time points had an advantage over models created with sparse time points or the single largest data set when applied to the external validation samples (Fig. 5C; Fig. S10). We also used eXtreme Gradient Boosting (XGB),

a boosting algorithm which serves the same feature selection and prediction functions as the bagging algorithms, including RF. The XGB models also differed significantly in prediction performance for early and late samples (Table S11). Together, these results suggest that pig age is associated with the gut microbiota and can be predicted using microbiota age, especially in pigs less than 80 days old.

## The dynamic core microbiota

The core microbiota refers to a group of consistent microbial taxa that are present across multiple hosts. The concept of a "core" microbiota in the swine gut is intriguing because it could lead to the identification of possible targets for dietary or pharmacological interventions in swine production systems. We chose to define the core microbiota as taxa present in at least 90% of all samples, an arbitrary but widely accepted threshold (19, 38, 39). Twenty-four genera were present in at least 90% of all samples, of which 17 genera belonged to the phylum *Bacillota*; 5 were members of the *Bacteroidota* phylum; and 2 belonged to the *Pseudomonadota* (Table S12).

Next, we sought to define stage-specific core microbiota groups that are present in at least 90% of samples in each growth stage. The genera *Lactobacillus*, *Clostridium sensu stricto 1*, *Christensenellaceae R-7 group*, (*Eubacterium*) *coprostanoligenes* group, *Parabacteroides*, *Desulfovibrio*, *Blautia*, *Lachnospiraceae UCG-008*, *Lachnoclostridium*, and *Phascolarctobacterium* were shared in the four stage-specific core groups (Fig. S11). Next, we loosened the threshold to 50% to identify a lifelong core microbiota. The core genera that are present in at least 50% of the samples from all time points were *Escherichia*, *Streptococcus*, *Lactobacillus*, *Clostridium sensu stricto 1*, *Bacteroides*, *Christensenellaceae R-7 group*, *Lachnoclostridium*, and *Faecalibacterium*. Overall, our results extended previous definitions of the pig core microbiota (19, 38).

## DISCUSSION

Swine-associated microbial communities can have strong positive and negative effects on pig disease and health (40), nutrition (41), growth performance (42), physiological development (43, 44), antimicrobial resistance (45, 46), as well as on the environment (47). Generally speaking, the swine production cycle can be divided into four phases. The suckling phase is the period from birth to weaning (approximately 3 weeks), during which the pig's diet is largely sow's milk. During the weaning or nursery (6–8 weeks), growing, to finishing (16–17 weeks) phases, pigs receive a solid diet with increasing fiber content and decreasing protein content. Understanding the dynamics of microbial communities in the swine gut is therefore an important aspect for swine production and the spatial-temporal progression of the swine gut microbiota, which have been intensively discussed recently (48, 49).

The natural variations and individuality of the gut colonization process have been well documented, and therefore, it is challenging to define a microbial developmental pattern that is applicable to all conditions (16, 25). If experimental observations are limited to a single data set which may not be representative, the impact of ecological factors driving community assembly will be overestimated (35). In this study, with over 3,000 longitudinal fecal samples from pigs spanning four continents, we analyzed the swine gut microbiota with the largest sample size to date and sought to identify a generalized gut microbiota developmental pattern in pigs. The focus on young piglets makes sense because most health problems in pigs arise during the suckling and weaning stages (50–52).

We were able to predict pig age as a function of OTU relative abundance using machine learning approaches. Both RF and XGB algorithms showed strong predictive capacity and accurately predicted the microbiota age of pigs (up to 80 days of age) in the external validation data set, despite not being used to train the models. Those OTUs identified from the RF models either decreased or increased in relative abundance over time, indicating their marker function. Since intestinal microbiome development is a continuous process, future studies could benefit from comparison to a continuous

reference (like our RF model) rather than snapshot comparisons to a treatment group. This approach has been adopted in human microbiome studies, and it has been reported to be important for human health (53–57). For example, the gut microbiota of malnourished Bangladeshi children was underdeveloped compared to healthy children (53), and the age-discriminatory OTUs from healthy human gut samples identified using RF models showed preventative effects on growth restriction in gnotobiotic mice colonized with the gut microbiota from malnourished children (55).

Unlike conventional meta-analyses, which assign effect sizes to each study and estimate the summary effect using a fixed effects model or random effects model (58), we performed a *de novo* analysis (IPD meta-analysis), i.e., re-analysis of the raw sequencing data (59). In an IPD meta-analysis, the original data from all studies are collected and combined into one data set. The specificity of the IPD meta-analysis approach makes it difficult to quantify the heterogeneity and publication bias, especially for time-series analysis (38, 60). In the current study, we acknowledge the presence of publication bias but believe this was minimized as all of the original studies were observational and most of our analysis was based on control samples from untreated pigs. We used a closed-reference OTU picking approach to reduce heterogeneity by incorporating all data sets into the same analysis pipeline and by matching all sequences against the SILVA database, which has commonly been used in microbiome IPD meta-analysis (59, 61–64).

Previous studies conducted with coarse time intervals or a limited time span have revealed that the composition of the gut microbiota continually changes during pig growth (60). Moreover, conclusions drawn from single data sets are challenging to generalize. Our high-resolution sampling analysis enabled us to characterize the developmental patterns of the gut microbiota over time in great detail and indicated that the recruitment of microbes is age sensitive. We demonstrated that there are temporal shifts in microbial composition during the life span of production pigs. During this period, there were more phyla and genera that increased in relative abundance in the gut than decreased. These results reinforce the observation of increased microbial diversity and richness with pig age.

The gut microbiota developed over the course of the first 30–50 days post-natal but then stabilized in composition thereafter. On the other hand, the gut microbiota in suckling pigs changed rapidly, although they consume the same liquid diet from their mother. This might be because of the development of the piglets' digestive tract physiology (e.g., increasingly anaerobic) and the change of the sow's milk composition over time (65). Another explanation could be the introduction of solid feed (creep feeding) in the last part of the suckling phase in some studies (12).

The concept of enterotype has been highly discussed and debated in human microbiome clustering analysis (66, 67). Researchers have argued that enterotypes are not discrete states that separate individuals but rather represent continuous gradients of dominating taxa between samples (68). Our DMM clustering results show that clusters are more like gradients of dominant taxa rather than discrete clusters of three or four enterotypes. Depending on the rarefaction choice we made, at least 10 distinct clusters were determined for the swine fecal microbiota over time. We observed an age-related pattern in the clusters with the transition stabilizing with age, indicating maturation of the swine gut microbiota. Because of study heterogeneity, the batch effect is unavoidable in meta-analyses. However, during the clustering procedure, no study-specific cluster was observed, demonstrating that clustering is independent of study.

Future studies should focus on the development of the swine microbiome over time, rather than simply time-matched treatment comparisons. It would be intriguing to determine if the bacterial species represented by the age-discriminatory OTUs we identified from pigs (i.e., as a probiotic treatment) could contribute to pig recovery following biotic or abiotic challenges that cause developmental delays. We observed increasing trends in the relative abundance of most age-discriminatory OTUs, indicating limited-time persistence of early colonizers. The early colonizers may be due to milk

selectivity and the later colonizers selected by the introduction of solid feed. The early colonizing microbiota showed immune activation function in humans and may exert a lifelong impact on host health (69, 70). It would also be interesting to investigate whether bacteria isolates represented by the early colonizing OTUs have an influence on the immune system and microbiota assembly in pigs.

When assessing certain microbiota-derived interventions, it is important to focus on their dynamic influence rather than their cross-sectional impact. Previously, age was considered a key factor in swine gut microbiome development, but there was no clear consensus on how the microbial community develops over time in swine. The lack of universal developmental patterns limits our understanding of the swine gut microbiome and the application of microbiota-derived therapies or growth promoters. We filled this knowledge gap on the dynamics of the swine gut microbiota using continuous data by describing the microbial community with extensive data sets to determine that the swine gut microbiota undergoes predictable community succession, at least until 80 days of age, after which additional characterization of the microbiome is needed. The microbial community changes rapidly in the early stage followed by a slower succession as pigs age. The identified microbial biomarkers in each stage may play an important role in animal gut homeostasis and need to be further explored.

## MATERIALS AND METHODS

### Data acquisition

Sequence data were obtained from 14 publications that included longitudinal sampling of the commercial pig fecal microbiota over at least three time points (Fig. 1). To avoid the potential bias associated with the large variation in sequencing depth observed in historical data sets (38, 60), only papers published from 2019 to 2021 were considered. A total of 3,313 samples including 2,260 control samples from untreated pigs were found by entering the keywords "pig microbiome," "swine microbiome," "pig microbial community," "swine microbial community," "pig microbiota," or "swine microbiota" into the Web of Science (https://www.webofscience.com/), PubMed (https://pubmed.ncbi.nlm.nih.gov/), and the Sequence Read Archive (https://www.ncbi.nlm.nih.gov/sra) databases in January 2022.

Metadata are included in Table S1 and include country, pig breed, DNA extraction approach, PCR primers used, and sampling time. Only control samples from untreated pigs were used to determine the microbial community succession independent of treatment effects, unless otherwise stated. Time points with a sample size exceeding 50 were resampled to 50 samples to account for bias associated with the non-uniform distribution of samples between studies, unless otherwise stated (Fig. S1).

### Sequence processing

PCR primers were found as a part of the sequence data in six data sets, and Cutadapt (version 3.4) was used to remove the primers from these sequences (71). Then, raw 16S rRNA gene amplicon sequence reads were imported and analyzed using QIIME2 (version 2021.11) (72). Here, we introduce the first QIIME2-based closed-reference OTU picking pipeline for meta-analysis (File S1). Previous meta-analyses have only used the QIIME1 script pick_closed_reference_otus.py, which is no longer being maintained (60, 61, 73–75). Briefly, paired-end reads were joined using the join-pairs method in the q2-vsearch plugin with a minimum overlap of 20 bp (76). Only forward reads were considered when >50% of the reads could not be merged, a threshold that has been used previously (59). Next, single-end and merged paired-end reads were quality filtered by the q2-quality-filter $q$-score function, with a minimum quality score threshold of 20 (77).

After dereplication with the dereplicate-sequence command, the quality-filtered reads were clustered into OTUs using the cluster-features-closed-reference function in the q2-vsearch plugin. Closed-reference clustering was performed at 97% identity

against the SILVA (132 QIIME release) 97% OTU reference database (78). Reads that failed to cluster with a sequence from the reference database at 97% similarity were discarded.

A total of 94,467,293 quality-filtered sequences were obtained and clustered into 35,929 OTUs after closed-reference OTU picking. The closed-reference OTU picking workflow is a database-dependent strategy that clusters sequences into OTUs and assigns taxonomy to reference sequences using a pre-existing collection of reference sequences with known taxonomy. Lastly, we used *de novo* chimera checking with the UCHIME-*de novo* command and excluded sequences containing chimeras using the q2-feature-table plugin with the filter-features command.

## Diversity analysis

Alpha and beta diversity analyses were conducted in QIIME2 using the q2-diversity plugin with core-metrics-phylogenetic command. In order to keep a reasonable number of samples for further analysis, samples were rarefied to 7,550 reads, which retained 90% of all samples, a sampling depth sufficient to observe differences between samples (79), especially when considering the differences between all studies included in the meta-analysis. The Shannon diversity index (80) and the number of observed OTUs were used as quantitative and qualitative measurements of microbial community diversity and richness, respectively. Faith's phylogenetic diversity (81) was calculated to determine phylogenetic diversity in the microbial communities, and Pielou's evenness (82) was used for community evenness evaluation. For beta diversity, Jaccard similarity (83), Bray-Curtis dissimilarity (84), and weighted and unweighted UniFrac distances (85) were calculated, and were visualized by PCoA plots using ggplot2 (86).

## DMM clustering

To determine the key phases of microbiota progression, we used DMM clustering as an unsupervised method to bin samples based on the structure of the microbial community (36). The lowest Laplace approximation given to the negative log model evidence was used to estimate the optimal number of clusters. Briefly, the DMM modeling was performed using the R package "DirichletMultinomial (version 1.34.0)" with the raw and rarefied OTU tables as well as the genus level feature table. The five OTUs contributing the most to each DMM cluster were determined based on the strength value given by the default output.

## Random forest regression

We developed regression models for the gut microbiota maturation using RF. The RF models can be used to select microbial community features that are age discriminatory, as well as to predict the microbiota developmental status (87). The relative abundance of OTUs without rarefaction was regressed against their chronological age according to the default parameters of the RF algorithm (R package "randomForest [version 4.6–14]"; ntree = 10,000, mtry = $P$/3, where $P$ is the number of input OTUs). Data sets from two publications were randomly selected as external validation samples (229 samples), and the remaining samples from the other 12 publications were divided into a training set (1,150 samples) and a test set (286 samples) based on the pig age using the "reateData-Partition" function in the "caret (version 6.0-90)" package. The "rfcv" function with 10-fold cross-validation was used to estimate the optimal number of top-ranking age-discriminatory OTUs required for prediction. To test the quality of the RF models created using varying amounts of training data, we also created RF models based on the single largest data set (37) and a data set with 10 randomly selected time points (days 7, 21, 35, 49, 61, 84, 99, 112, 130, and 146). The performance of these RF models was evaluated by applying them to the external validation samples.

As a complement to the RF models, we also applied the XGB algorithm to regress the microbiota age (88). We used the same data sets to perform the XGB modeling in 20 iterations using the "xgb.train" function in the "xgboost (version 1.5.0.2)" package with

the following parameters: eta = 0.1; subsample = 0.8; nrounds = 2,000; and early_stopping_rounds = 200.

## Statistical analysis

All statistical analyses were performed in the R environment (version 4.1.1) (89). PERMANOVA was used to compare the beta diversity between groups using the "adonis" function in the "vegan (version 2.6-2)" package (90), and PERMDISP was used to analyze the data dispersion between groups using the "betadisper" function in the "vegan" package. The "lmer" function in the "lme4" package was used to examine changes over time in the longitudinal data using a linear mixed-effect model with study as the random effect (91). The Kruskal-Wallis test was used to assess changes in alpha diversity between DMM clusters, followed by a multiple comparison using the Wilcoxon rank-sum test with Benjamini-Hochberg correction for $P$ value adjustment due to multiple testing. The relative abundance of the five OTUs that contributed the most to each DMM cluster was analyzed using the Kruskal-Wallis test followed by Dunn's test for multiple comparisons with correction for multiple testing using Holm's method (92). Beta regression with study and pig age as regressors was run using the "betareg" function in the "BetaReg (version 3.1-4)" package (93). To evaluate the accuracy of the RF and XGB models, root mean squared error, mean absolute error, and $R^2$ were calculated with the "defaultSummary" function in the "caret" package. Linear models were performed using the "lm" function; analysis of variance was performed using the "aov" function, and Pearson's $r$ and Spearman's rho were calculated using the "cor.test" function from the "stats" package. The results were considered significant when $P$ values were <0.05.

## ACKNOWLEDGMENTS

We thank Drs. Michiel Kleerebezem and Raka Choudhury from the Department of Animal Sciences, Wageningen University & Research, and Dr. Olivier Zemb from GenPhySE, Université de Toulouse, INRA, for their kind support for the metadata. We thank Dr. Greg Caporaso from Northern Arizona University for helpful discussions in the preparation of the analysis pipeline.

This research received no specific grant from any funding agency in the public, commercial, or not-for-profit sectors.

W.D. and T.A.J. conceived the study; W.D. collected the data sets and performed the data analysis; W.D., T.A.J., D.B.H., and N.R. discussed the results; W.D. wrote the manuscript; and T.A.J., D.B.H., and N.R. edited the manuscript. All authors read and approved the final manuscript.

## AUTHOR AFFILIATIONS

[1]Department of Animal Sciences, Purdue University, West Lafayette, Indiana, USA
[2]Department of Pathobiology, University of Guelph, Guelph, Ontario, Canada
[3]Lacombe Research and Development Centre, Agriculture and Agri-Food Canada, Lacombe, Alberta, Canada

## AUTHOR ORCIDs

Wenxuan Dong  http://orcid.org/0000-0002-0867-8583
Nicole Ricker  https://orcid.org/0000-0001-5706-5399
Devin B. Holman  https://orcid.org/0000-0001-5306-3732
Timothy A. Johnson  http://orcid.org/0000-0001-8204-547X

## FUNDING

| Funder | Grant(s) | Author(s) |
|---|---|---|
| Purdue University (PU) | | Wenxuan Dong |

| Funder | Grant(s) | Author(s) |
|--------|----------|-----------|
| | | Timothy A. Johnson |

## AUTHOR CONTRIBUTIONS

Wenxuan Dong, Conceptualization, data curation, formal analysis, investigation, methodology, software, visualization, writing - original draft, Writing – review and editing | Nicole Ricker, investigation, Writing – review and editing | Devin B. Holman, investigation, Writing – review and editing | Timothy A. Johnson, Conceptualization, Project administration, Supervision, Writing – review and editing

## DATA AVAILABILITY

All data sets analyzed in this study are available from public databases as identified in Table S1. The QIIME2 script and R scripts are available on GitHub at https://github.com/dong316/meta.

## ADDITIONAL FILES

The following material is available online.

### Supplemental Material

**Supplemental figure legends (Spectrum01722-23-s0001.docx).** Legends for Fig. S1 to S11.
**Supplemental figures (Spectrum01722-23-s0002.pdf).** Fig. S1 to S11.
**Supplemental text (Spectrum01722-23-s0003.txt).** Qiime2 commands for closed reference clustering and analysis.
**Supplemental tables (Spectrum01722-23-s0004.xlsx).** Tables S1 to S12.

### Open Peer Review

**PEER REVIEW HISTORY (review-history.pdf).** An accounting of the reviewer comments and feedback.

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
