## [Reviewer comments · Microbiology Spectrum]

Microbiology Spectrum

Meta-analysis reveals the predictable dynamic development of the gut microbiota in commercial pigs

Wenxuan Dong, Nicole Ricker, Devin Holman, and Timothy Johnson

Corresponding Author(s): Timothy Johnson, Purdue University

Review Timeline:

Submission Date:	May 10, 2023
Editorial Decision:	June 22, 2023
Revision Received:	July 21, 2023
Editorial Decision:	August 11, 2023
Revision Received:	August 18, 2023
Accepted:	August 24, 2023

Editor: Stephan Schmitz-Esser

Reviewer(s): Disclosure of reviewer identity is with reference to reviewer comments included in decision letter(s). The following individuals involved in review of your submission have agreed to reveal their identity: Michiel Van de Vliet (Reviewer #1); Hao-Yu Liu (Reviewer #2)

Transaction Report:

DOI: <https://doi.org/10.1128/spectrum.01722-23>

June 22, 2023

Dr. Timothy A Johnson
Purdue University
Department of Animal Sciences
Purdue University
Creighton Hall of Animal Sciences
West Lafayette, Indiana 47906

Re: Spectrum01722-23 (Meta-analysis reveals the predictable dynamic development of the gut microbiome in commercial pigs)

Dear Dr. Timothy A Johnson:

Thank you for submitting your manuscript to Microbiology Spectrum. Your manuscript has been reviewed by two experts in the field. Please see their comments below. When submitting the revised version of your paper, please provide (1) point-by-point responses to the issues raised by the reviewers as file type "Response to Reviewers," not in your cover letter, and (2) a PDF file that indicates the changes from the original submission (by highlighting or underlining the changes) as file type "Marked Up Manuscript - For Review Only". Please use this link to submit your revised manuscript - we strongly recommend that you submit your paper within the next 60 days or reach out to me. Detailed instructions on submitting your revised paper are below.

Link Not Available

Sincerely,

Stephan Schmitz-Esser

Journals Department
Reviewer comments:

Reviewer #1 (Comments for the Author):

To the Authors

In this paper 'Meta-analysis reveals the predictable dynamic development of the gut microbiome in commercial pigs', Dong and colleagues present their re-analysis of published datasets on pig gut microbiome development. Using their RF model, they predict the age of the piglets based on their gut microbiota. Overall, the paper is clearly written and contains important messages for the scientific community, but some points that need further clarification are raised below:

- Overall comment

- o The names of the phyla are not according to the most recent guidelines. Please adapt this throughout the manuscript.
- DMM clustering
 - o DMM clustering is performed using the control and treated pigs. The treatments appear often to have a maturing effect on the gut microbiota community. Also, weaning day differs from study to study and also weaning is associated with the maturation of the gut microbiota, as described by . Have you assessed whether both factors influenced the DMM clustering and/or introduced artificial clusters in the transition area? Please clarify.
 - Figure 1:
 - o Figure 1C is not referred to in the text. Please also add a timeline indicating the growth stages of the pig and from which day to which day these stages go, to further clarify the figure. Since days and growth stages are used interchangeably in the article, this illustration will make it easier to follow.
 - o The pie charts in figure 1B overlap. Please adapt this.
 - Figure 3
 - o Figure 3 is counterintuitive now as it seems that early-day samples cluster better together than late-day samples. Would it be possible to visualize the clusters (to the example of figure S7) to improve this? (suckling vs finishing)
 - o Also, there is a typo in the figure description (Figure 3 and S6). Panel A-B and C-D need to be switched in the description of the figures.
 - Figure 4
 - o Figure 4B is rather complex since many colors are used and not everything on the figure has an added value. As clustering of the phyla is not further discussed, this can be left out. To further clarify the figure, the names of the cluster (letters) on the other hand could be added on top and on the left to their respective color for legibility.

Reviewer #2 (Comments for the Author):

In the manuscript "Meta-analysis reveals the predictable dynamic development of the gut microbiome in commercial pigs", Wenxuan Dong, et al. used a total of 3,313 fecal microbiota sampling from over 349 pigs covering 60 time points (from birth to market age) from 14 publications to make meta-analysis. The authors' detailed characterization of gut microbiome through meta-analysis enables an understanding of the dynamics of microbial community succession, as well as transient and natural variation between timepoints and animals.

Major

1. Please be clear with the concept of "microbiome", it is defined as ""a characteristic microbial community occupying a reasonably well-defined habitat which has distinct physio-chemical properties." The term thus not only refers to the microbial community composition, but also encompasses their activity and function. In the current study, it is generally the microbiota analyzed than microbiome. In addition, the manuscript mentioned "microbial taxa were recruited" several times, I assume it was talking about bacterial succession, but how exactly microbes are "recruited"? From where? Please explain.
2. The significance of the paper wasn't clear. What is the new? The dynamics of microbial community succession? Yes, however, it is more or less known. "Potential microbiota biomarkers"? Please specify what biomarkers are identified. The manuscript also mentioned that they identified core microbiota. These core microbiotas are shared and constant in life time of pigs. Isn't that they are the opposite of the dynamics, and how they can be manipulated? Please elaborate.
3. The link between pig study and human research wasn't solid. In introduction, it was mentioned that in human study, high-frequency sampling and analysis were conducted. Therefore, it is necessary to do the same in pig study. So, could the already done analysis apply directly in the current study, or the methodology developed in the current study could promote human study? Please explain.
4. Table S1 and Figure 1 should be combined. Figure 1 A doesn't show sampling frequency clearly. Figure 1C (plus legend) does not explain the time range and the scheme clearly.
5. The manuscript accurately predicted the microbiota age of pigs (up to 80 days of age) in the external validation dataset, what does "80 days " mean and why did you choose this point in time?
6. The manuscript used "DMM" model to predict whether the gut microbiota clusters with age, but the gut microbiota can be influenced by many factors, such as diet and environment, so how to avoid interference from these factors ?

Minor

1. Figure 1 should be revised, the pie chart in Figure 1B is too tightly laid out and should be properly spaced
2. Figure 2 should be revised, the layout of the images is too messy, the P value in the legend of Figure 2A should be italicized (none of the significant P-value in the article are in italics), the font of the legend on the right side of Figure 2B is squeezed together, and the fonts of the X and Y axes are not very clear in grey, which is not consistent with the font of the images.
3. Figure 4 should be revised, why is there a letter "C" in the background at the bottom of Figure 4B, and the placement of the legend is a bit confusing.

Staff Comments:

Preparing Revision Guidelines

Please return the manuscript within 60 days; if you cannot complete the modification within this time period, please contact me. If you do not wish to modify the manuscript and prefer to submit it to another journal, please notify me of your decision immediately so that the manuscript may be formally withdrawn from consideration by Microbiology Spectrum.

To the Authors

In this paper 'Meta-analysis reveals the predictable dynamic development of the gut microbiome in commercial pigs', Dong and colleagues present their re-analysis of published datasets on pig gut microbiome development. Using their RF model, they predict the age of the piglets based on their gut microbiota. Overall, the paper is clearly written and contains important messages for the scientific community, but some points that need further clarification are raised below:

- **Overall comment**
 - The names of the phyla are not according to the most recent guidelines. Please adapt this throughout the manuscript.
- **DMM clustering**
 - DMM clustering is performed using the control and treated pigs. The treatments appear often to have a maturing effect on the gut microbiota community. Also, weaning day differs from study to study and also weaning is associated with the maturation of the gut microbiota, as described by . Have you assessed whether both factors influenced the DMM clustering and/or introduced artificial clusters in the transition area? Please clarify.
- **Figure 1:**
 - Figure 1C is not referred to in the text. Please also add a timeline indicating the growth stages of the pig and from which day to which day these stages go, to further clarify the figure. Since days and growth stages are used interchangeably in the article, this illustration will make it easier to follow.
 - The pie charts in figure 1B overlap. Please adapt this.
- **Figure 3**
 - Figure 3 is counterintuitive now as it seems that early-day samples cluster better together than late-day samples. Would it be possible to visualize the clusters (to the example of figure S7) to improve this? (suckling vs finishing)
 - Also, there is a typo in the figure description (Figure 3 and S6). Panel A-B and C-D need to be switched in the description of the figures.
- **Figure 4**
 - Figure 4B is rather complex since many colors are used and not everything on the figure has an added value. As clustering of the phyla is not further discussed, this can be left out. To further clarify the figure, the names of the cluster (letters) on the other hand could be added on top and on the left to their respective color for legibility.

We would like to thank the reviewers for their helpful critiques of our manuscript. We have utilized and addressed all comments to improve the quality of our manuscript. Please see below for an item-by-item response.

Reviewer #1 (Comments for the Author):

To the Authors

In this paper 'Meta-analysis reveals the predictable dynamic development of the gut microbiome in commercial pigs', Dong and colleagues present their re-analysis of published datasets on pig gut microbiome development. Using their RF model, they predict the age of the piglets based on their gut microbiota. Overall, the paper is clearly written and contains important messages for the scientific community, but some points that need further clarification are raised below:

- Overall comment: The names of the phyla are not according to the most recent guidelines. Please adapt this throughout the manuscript.

Response: We gratefully appreciate for your valuable suggestion. We have changed the names of the phyla according to the most recent guidelines. See L222-223, Figure 2.

- DMM clustering is performed using the control and treated pigs. The treatments appear often to have a maturing effect on the gut microbiota community. Also, weaning day differs from study to study and also weaning is associated with the maturation of the gut microbiota, as described by . Have you assessed whether both factors influenced the DMM clustering and/or introduced artificial clusters in the transition area? Please clarify.

Response: Thank you for your comments. The objective of DMM analysis was to determine the age-dependent pattern of the gut microbiota. We hoped to improve the robustness of clustering selections by adding samples from treated pigs, which could provide more biological variation (1,2). We completed additional analysis to determine if known animal treatments caused any specific clustering patterns, and we found there were no treatment-specific clusters, so we only focused on the control samples downstream as in other analyses in this manuscript. We added this information to the results:

Line 179-182: “Various animal treatments (fecal microbiota transplant, antibiotic treatment, and feed restriction) were equally represented in the DMM clusters, so we concluded that age was the primary factor in community clustering patterns.”

References

1. Stewart, Christopher J., et al. "Temporal development of the gut microbiome in early childhood from the TEDDY study." *Nature* 562.7728 (2018): 583-588.
2. Holmes, I., Harris, K. & Quince, C. Dirichlet multinomial mixtures: generative models for microbial metagenomics. *PLoS ONE* 7, e30126 (2012).

Figure 1: Figure 1C is not referred to in the text. Please also add a timeline indicating the growth stages of the pig and from which day to which day these stages go, to further clarify the figure. Since days and growth stages are used interchangeably in the article, this illustration will make it easier to follow. The pie charts in figure 1B overlap. Please adapt this.

Response: Thank you for your comments. We decided to remove Figure 1C because it contributes little to the presentation of the research work in this manuscript, and may actually cause confusion due to its brevity. Although days and growth stages can be used interchangeably, different feeding regimes were used in different papers re-analyzed here. There is no clearly separation of growth stages based on the age of pigs world-wide. Instead, we added a brief introduction of the time span of the swine production cycle in the first paragraph of discussion, which will provide more information than was conveyed in Figure 1C and can help readers have a better understanding.. Also, Figure 2B has been revised to avoid overlaps.

- Figure 3: Figure 3 is counterintuitive now as it seems that early-day samples cluster better together than late-day samples. Would it be possible to visualize the clusters (to the example of figure S7) to improve this? (suckling vs finishing). Also, there is a typo in the figure description (Figure 3 and S6). Panel A-B and C-D need to be switched in the description of the figures.

Response: Thank you for your nice suggestions. The reviewer is correct that there appears to be some clustering of the samples from suckling animals in Fig. 3. We improved the aesthetics in Fig. 3 to visualize the community shifts more clearly. Panels A and B (Figures 3 and S6) now include 95% confidence interval ellipses for each stage. The ellipses indicate that late stages (growing and finishing) are generally smaller than early stages (suckling and weaning) according to all four measures of beta diversity. Suckling samples have more dispersal in PCo1 axis and finishing samples have more dispersal in PCo2 axis. We also increased the number of colors in panels C and D in efforts to more clearly indicate different ages. It seems the community shift occurs in large extent by day 30, indicated by yellow points.

Regarding if the clustering pattern is counter-intuitive. We assume the reviewer is saying this because of the patterns observed in the distance matrices (Fig. 3E-F). It is important to remember when comparing the PCoAs and the distance matrix, that the distance matrix represents the actual pairwise distances, while the PCoA only conveys a portion of the variability of the distance matrix (about 17% in the case of Bray-Curtis). Thus the relationships between samples are more accurate in the distance matrix. Given the number of samples in this dataset, we decided to include the distance matrix to convey more accurate distances. In the distance matrix, there is a limited degree of stability between

suckling samples (there is some white and light blue in the lower left corner). However, we feel that this degree of stability is low. It is an interesting pattern and one we want to explore in future experiments.

The typo in the figure legends has been fixed. We switched the description of panels (AB) and (CD).

- Figure 4: Figure 4B is rather complex since many colors are used and not everything on the figure has an added value. As clustering of the phyla is not further discussed, this can be left out. To further clarify the figure, the names of the cluster (letters) on the other hand could be added on top and on the left to their respective color for legibility.

Response: Thank you for your nice suggestions. Figure 4 has been revised to reduce its complexity. Unnecessary legends has been removed. We removed the phylum legends since it is not further discussed. We also removed the DMM legend by adding the names on the top and on the left to their respective colors. We also removed the letter “C” in the background at the bottom of Figure 4B.

Reviewer #2 (Comments for the Author):

In the manuscript "Meta-analysis reveals the predictable dynamic development of the gut microbiome in commercial pigs", Wenxuan Dong, et al. used a total of 3,313 fecal microbiota sampling from over 349 pigs covering 60 time points (from birth to market age) from 14 publications to make meta-analysis. The authors' detailed characterization of gut microbiome through meta-analysis enables an understanding of the dynamics of microbial community succession, as well as transient and natural variation between timepoints and animals.

Major

1. Please be clear with the concept of "microbiome", it is defined as ""a characteristic microbial community occupying a reasonably well-defined habitat which has distinct physio-chemical properties." The term thus not only refers to the microbial community composition, but also encompasses their activity and function. In the current study, it is generally the microbiota analyzed than microbiome. In addition, the manuscript mentioned "microbial taxa were recruited" several times, I assume it was talking about bacterial succession, but how exactly microbes are "recruited"? From where? Please explain.

Response: Thank you for your thoughtful comments. We have changed the term "microbiome" to "microbiota" when it refers to only the microbial component of the microbiome but not to other features of the gut environment. We also replaced the word "recruited" to "enriched" since it is describing the relative abundances of the microbial taxa. (L25, 133)

2. The significance of the paper wasn't clear. What is the new? The dynamics of microbial community succession? Yes, however, it is more or less known. "Potential microbiota biomarkers"? Please specify what biomarkers are identified. The manuscript also mentioned that they identified core microbiota. These core microbiotas are shared and constant in life time of pigs. Isn't that they are the opposite of the dynamics, and how they can be manipulated? Please elaborate.

Response: Thank you for your comments. We have tried to make the significance more clear in the last sentence of the abstract,

"Despite differences in animal breeds and management, external validation showed this model could be generalized to future microbiome studies conducted in suckling and weaning pigs to quantify community succession."

Primarily, we think the most significant piece of this work is the generalizable pattern of swine microbial community succession, that we can quantify community succession by determining microbiota age, and there are bacterial

biomarkers of specific ages, as identified by random forest analysis.

For individual studies, it is challenging to quantify the microbial development using single data set with limited sample size and time points. Our meta-analysis provided new facets of the developmental swine microbiota using the largest data set and concluded that generalized age-dependent microbial developmental patterns exist, independent of specific animal management practices.

The presence of core microbiotas does not indicate that the community is not dynamic. The core microbiotas and the dynamic nature of the community reflect two important aspects of the swine gut microbiota. The core microbiotas reflect the stability of the gut microbiota. The core microbiotas are important in maintaining the homeostasis of gut microbiome. They provide potential targets in future manipulation. However, dynamics, and we would add, predictable dynamics, of the community reflects the path of the microbiota to a relatively stable state. One way to manipulate the gut microbiome would be to regulate the host-microbe and microbe-microbe interactions. The dynamics enable us to better explore the above interactions and achieve successful manipulations.

3. The link between pig study and human research wasn't solid. In introduction, it was mentioned that in human study, high-frequency sampling and analysis were conducted. Therefore, it is necessary to do the same in pig study. So, could the already done analysis apply directly in the current study, or the methodology developed in the current study could promote human study? Please explain.

Response: Thank you for your comments. 1. The analysis already completed in humans have provided numerous in-depth insights in disease diagnosis and treatment, and health interventions. Swine production is like-wise dealing with issues related to gut health, particularly those brought on by dysbiosis of the gut microbiota. We anticipate that a more precise, human-like comprehension will enable us to make advancements in the swine industry. 2. The human studies revealed the importance of large sample sizes and repeated measurements in gut microbiome research, which triggered us to do the current analysis. 3. We borrowed some analysis approaches from humans to help us better understand the development of swine gut microbiota, e.g. microbiota age, and DMM analysis.

The first QIIME2 based closed-reference OTU picking pipeline in this paper could be directly used to facilitate human studies.

4. Table S1 and Figure 1 should be combined. Figure 1 A doesn't show sampling frequency clearly. Figure 1C (plus legend) does not explain the time range and the scheme clearly.

Response: Thank you for your comments. Table S1 serves as supplementary material for Figure 1. Figure 1 generally introduces the locations, sampling days, sample sizes, and sequencing information. Readers will know why we complete the next steps of the analysis based on knowledge on Figure 1, e.g., why

sub-sampling was completed to avoid bias introduced by uneven sample sizes, why we choose closed reference OTU picking based on sequence region, how our data sets represents global sampling, etc. However, readers may be more interested in one specific paper, or the genetic background of pigs used in this analysis, etc. Readers may get more information from the Table S1 if they would like to know anything else that is not directly associated with the current analysis.

We decided to remove Figure 1C because it contributes little to the presentation of the research work in this manuscript, and may actually cause confusion due to its brevity. Although days and growth stages can be used interchangeably, different feeding regimes were used in different papers re-analyzed here. There is no clearly separation of growth stages based on the age of pigs world-wide. Instead, we added a brief introduction of the time span of the swine production cycle in the first paragraph of discussion, which will provide more information than Figure 1C and can help readers have a better understanding.

5. The manuscript accurately predicted the microbiota age of pigs (up to 80 days of age) in the external validation dataset, what does "80 days " mean and why did you choose this point in time?

Response: Thank you for your comments. Samples with an age of less than 80 days represent 85% (2818 out of 3313) of all samples and 84% (1910 out of 2261) of control samples collected in the current analysis. Based on most feeding regimes in the collected data, samples with ages less than 80 days covered both suckling, weaning, and part of the growing stages of pigs' life. A previous study also observed that after postnatal day 77, microbiota age is no longer significantly positively correlated with chronological age (Chang et al, PNAS, 2021, Figure S7). The choice of 80 days was based on our observation that the correlation between microbiota age and chronological age was reduced when samples older than 80 days were included in the correlation. The exact day as to when the prediction is accurate is not as important as understanding the performance of the random forest regression model was robust in younger pigs.

Chang, Hao-Wei, et al. "Gut microbiome contributions to altered metabolism in a pig model of undernutrition." Proceedings of the National Academy of Sciences 118.21 (2021): e2024446118.

6. The manuscript used "DMM" model to predict whether the gut microbiota clusters with age, but the gut microbiota can be influenced by many factors, such as diet and environment, so how to avoid interference from these factors?

Response: Thank you for your comments. The objective of DMM analysis was to determine the age-dependent pattern of the gut microbiota. We hoped to improve the robustness of clustering selections by adding samples from treated pigs, which could provide more biological variation (1,2). We completed additional analysis to determine if known animal treatments caused any specific clustering

patterns, and we found there were no treatment-specific clusters, so we only focused on the control samples downstream as in other analyses in this manuscript. We added this information to the results:

Line 179-182: “Various animal treatments (fecal microbiota transplant, antibiotic treatment, and feed restriction) were equally represented in the DMM clusters, so we concluded that age was the primary factor in community clustering patterns.”

1. Stewart, Christopher J., et al. "Temporal development of the gut microbiome in early childhood from the TEDDY study." *Nature* 562.7728 (2018): 583-588.

2. Holmes, I., Harris, K. & Quince, C. Dirichlet multinomial mixtures: generative models for microbial metagenomics. *PLoS ONE* 7, e30126 (2012).

Minor

- Figure 1 should be revised, the pie chart in Figure 1B is too tightly laid out and should be properly spaced

Response: Thank you for your nice suggestions. It has been revised. There are no overlaps after re-organization.

- Figure 2 should be revised, the layout of the images is too messy, the P value in the legend of Figure 2A should be italicized (none of the significant P-value in the article are in italics), the font of the legend on the right side of Figure 2B is squeezed together, and the fonts of the X and Y axes are not very clear in grey, which is not consistent with the font of the images.

Response: Thank you for your helpful suggestions. It has been revised. P values in this manuscript (including figures and the main text) has been italicized. See Fig.2, Fig.S2, Fig.S3, Fig.S5, Fig.S9, Fig.S10, L114, L198, L203.

The legend of Figure 2B has been revised to avoiding overlaps. The fonts of X and Y axes has been modified from grey to black.

- Figure 4 should be revised, why is there a letter "C" in the background at the

bottom of Figure 4B, and the placement of the legend is a bit confusing.

Response: Thank you for your helpful suggestions. Figure 4 has been revised to reduce its complexity. Unnecessary legends has been removed. We removed the phylum legends since it is not further discussed. We also removed the DMM legend by adding the names on the top and on the left to their respective colors. We also removed the letter “C” in the background at the bottom of Figure 4B.

August 11, 2023

Dr. Timothy A Johnson
Purdue University
Department of Animal Sciences
Purdue University
Creighton Hall of Animal Sciences
West Lafayette, Indiana 47906

Re: Spectrum01722-23R1 (Meta-analysis reveals the predictable dynamic development of the gut microbiota in commercial pigs)

Dear Dr. Timothy A Johnson:

Thank you for submitting your manuscript to Microbiology Spectrum. As you will see your paper is very close to acceptance. Please modify the manuscript along the lines recommended by reviewer #1. As these revisions are quite minor, I expect that you should be able to turn in the revised paper in less than 30 days, if not sooner. If your manuscript was reviewed, you will find the reviewers' comments below.

When submitting the revised version of your paper, please provide (1) point-by-point responses to the issues raised by the reviewers as file type "Response to Reviewers," not in your cover letter, and (2) a PDF file that indicates the changes from the original submission (by highlighting or underlining the changes) as file type "Marked Up Manuscript - For Review Only". Please use this link to submit your revised manuscript. Detailed instructions on submitting your revised paper are below.

Link Not Available

Sincerely,

Stephan Schmitz-Esser

Reviewer comments:

Reviewer #1 (Comments for the Author):

To the Authors

The overall comments raised before were addressed properly.

The extra analysis regarding the influence of treatments on DMM clustering was helpful. However, I would also add this analysis in Figure S8 and refer to it in the extra paragraph added in the text.

The adaptations to the figures have made them much clearer and easier to understand. Especially the addition of 95% confidence intervals and adjustments to the colors in Figure 3 have increased the intelligibility of the figure.

Reviewer #2 (Comments for the Author):

The authors have provided a detailed response to the comments and have revised the figures and tables accordingly. The study reveals changes of the porcine gut microbial community structure with age by analyzing a large amount of data and identifies potential microbial biomarkers that influence the development of the porcine gut microbiota, which can help to understand the mechanisms of microbiome assembly during host development.

Preparing Revision Guidelines

Please return the manuscript within 60 days; if you cannot complete the modification within this time period, please contact me. If you do not wish to modify the manuscript and prefer to submit it to another journal, please notify me of your decision immediately so that the manuscript may be formally withdrawn from consideration by Microbiology Spectrum.

To the Authors

The overall comments raised before were addressed properly.

The extra analysis regarding the influence of treatments on DMM clustering was helpful. However, I would also add this analysis in Figure S8 and refer to it in the extra paragraph added in the text.

The adaptations to the figures have made them much clearer and easier to understand. Especially the addition of 95% confidence intervals and adjustments to the colors in Figure 3 have increased the intelligibility of the figure.

We would like to thank the reviewers for their helpful critiques of our manuscript. We have utilized and addressed all comments to improve the quality of our manuscript. Please see below for an item-by-item response.

Reviewer #1 (Comments for the Author):

The overall comments raised before were addressed properly.

The extra analysis regarding the influence of treatments on DMM clustering was helpful. However, I would also add this analysis in Figure S8 and refer to it in the extra paragraph added in the text.

This was completed as suggested. Figure S8 now contains this extra analysis as panel Fig. S8E. Text was changed on lines 182 and 186, as well as the legend for Figure S8. Thank you to the reviewer for this suggestion as it strengthens the message of the paper.

The adaptations to the figures have made them much clearer and easier to understand. Especially the addition of 95% confidence intervals and adjustments to the colors in Figure 3 have increased the intelligibility of the figure.

Thank you for the supportive comments!

Reviewer #2 (Comments for the Author):

The authors have provided a detailed response to the comments and have revised the figures and tables accordingly. The study reveals changes of the porcine gut microbial community structure with age by analyzing a large amount of data and identifies potential microbial biomarkers that influence the development of the porcine gut microbiota, which can help to understand the mechanisms of microbiome assembly during host development.

Thank you for the supportive comments!

August 24, 2023

Dr. Timothy A Johnson
Purdue University
Department of Animal Sciences
Purdue University
Creighton Hall of Animal Sciences
West Lafayette, Indiana 47906

Re: Spectrum01722-23R2 (Meta-analysis reveals the predictable dynamic development of the gut microbiota in commercial pigs)

Dear Dr. Timothy A Johnson:

Your manuscript has been accepted, and I am forwarding it to the ASM Journals Department for publication. You will be notified when your proofs are ready to be viewed.

Sincerely,

Stephan Schmitz-Esser
Editor, Microbiology Spectrum
